# DINER: a Large Realistic Dataset for Evaluating Compositional Generalization

**Chengang Hu** and **Xiao Liu** and **Yansong Feng**[*]
Wangxuan Institute of Computer Technology, Peking University
{hcg,lxlisa,fengyansong}@pku.edu.cn

## Abstract

Most of the existing compositional generalization datasets are synthetically-generated, resulting in a lack of natural language variation. While there have been recent attempts to introduce non-synthetic datasets for compositional generalization, they suffer from either limited data scale or a lack of diversity in the forms of combinations. To better investigate compositional generalization with more linguistic phenomena and compositional diversity, we propose the DIsh NamE Recognition (DINER) task and create a large realistic Chinese dataset. Given a recipe instruction, models are required to recognize the dish name composed of diverse combinations of food, actions, and flavors. Our dataset consists of 3,811 dishes and 228,114 recipes, and involves plenty of linguistic phenomena such as anaphora, omission and ambiguity. We provide two strong baselines based on T5 (Raffel et al., 2020) and large language models (LLMs). This work contributes a challenging task, baseline methods to tackle the task, and insights into compositional generalization in the context of dish name recognition.

## 1 Introduction

Compositional generalization, the capacity to comprehend and create new combinations of observed components (Chomsky, 1956), has been shown to be an aspect of human intelligence both empirically (Fodor and Pylyshyn, 1988) and experimentally (Lake et al., 2019). Taking three recipes in Figure 1 as an example, by learning the instructions of cooking *braised pork (*红烧肉*)* and *marinated beef brisket (*卤牛腩*)*, we can acquire knowledge about components in dish names, like the use of *light soy sauce* and *dark soy sauce* usually indicates the action *braise*, and *beef brisket* is more likely to be a staple ingredient than *spices*. Given such knowledge, when reading the instructions of cooking *braised beef brisket (*红烧牛腩*)*, we can

---
[*] Corresponding author.

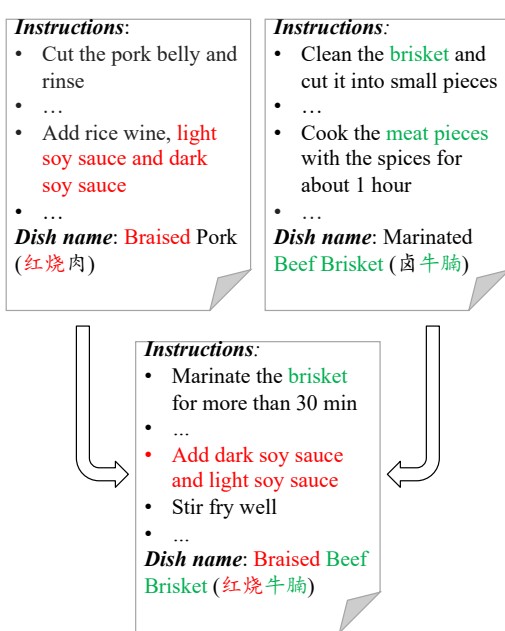

Figure 1: An example of compositional generalization in *dish name prediction*. Models are required to learn knowledge about *braised* and *beef brisket* and predict the composed dish name *braised beef brisket* given the corresponding instructions.

find the features corresponding to summarizing the staple ingredient *beef brisket* and abstract the action *braise*, thereby compose them into the name of this dish.

A bunch of synthetically-generated datasets have been created for assessing compositional generalization (Lake and Baroni, 2017; Bastings et al., 2018; Keysers et al., 2020; Kim and Linzen, 2020), and plain sequence-to-sequence(seq2seq) models exhibit significant out-of-distribution (OOD) compositional generalization performance loss comparing to in-distribution (ID) setting. While effective methods have been proposed to overcome the difficulty in OOD compositional generalization (Lake, 2019; Nye et al., 2020; Weißenhorn et al., 2022), most of them mainly focus on semantic parsing, where some important abilities like summarization

| Dataset | # of samples | generalization forms |
|---|---|---|
| GEOQUERY (Shaw et al., 2021) | 880 | 3 |
| SPIDER-SSP (Shaw et al., 2021) | 4,376 | 3 |
| SMCALFLOW-CS (Yin et al., 2021) | 28,054 | 2 |
| COUNTERFACTUAL (Liu et al., 2022) | 2,500 | 1 |
| DINER(ours) | 223,581 | 4 |

Table 1: Scale and diversity of non-synthetic compositional generalization datasets

and abstracting are not involved. Moreover, they are only evaluated on synthetic data, which is lack of natural language variation (Shaw et al., 2021) such as anaphora, omission and ambiguity (Keysers et al., 2020).

Recently, non-synthetic datasets have been proposed to evaluate compositional generalization on more realistic scenario (Shaw et al., 2021; Liu et al., 2022; Yin et al., 2021). However, as shown in Table 1, these datasets are either limited in data scale, which may lead to over-fitting and high variance in evaluation; or lack diversity of generalization forms (further explained in §2), which can result in biased models and limited generalization capabilities.

To address the aforementioned issues, we propose the DIsh NamE Recognition (DINER) task, and build large-scale Chinese dataset with a diverse collection of combinations. In this task, given the instructions of a recipe, models are required to predict the dish name[1], which is often composed of up to three kind of components: food, actions and flavor. As visualized in Figure 1, models will be tested on unseen dish names composed by observed components.

This task is not trivial even without compositional generalization setting. Firstly, it takes summarization and abstracting to figure out which ingredients, actions and flavors should be included in dish name. Secondly, the presence of anaphora, omission and ambiguity in recipes further increase the difficulty. We discuss such difficulty in §3.5 with a more comprehensive analysis.

Based on the XIACHUFANG Recipe Dataset (Liu et al., 2022), we collect 3,803 dish names that can be parsed into ingredients, actions and flavors, with 223,581 corresponding recipes. As shown in Table 3, the large data scale brings a rich collection of combination forms. Then we generate TMCD (Shaw et al., 2021) splits of the dish names for assessing compositional

generalization. In this approach, training and test sets are partitioned in a manner that maximizes the divergence of compound distribution. The TMCD splits result in 4 categories of generalization forms, thereby raise more challenges in our compositional generalization setting.

We develop baseline methods to solve the task, as existing compositional generalization models cannot fit in the dish name prediction task form. The first baseline, seq2seq trained T5, suffers from a significant ID-OOD performance gap. Inspired by the chain of thought prompting method (Wei et al., 2022), we propose to fine tune with compositional prompting (CP-FT). We first train a component model for predicting each kind of components, then we add the predictions of components to the input, and fine tune the model with the compositional prompt. To better fit the T5 model to the recipe domain, we also perform continual pretraining on a large-scale recipe corpus to equip models with domain knowledge. The proposed CP-FT method and continual pretraining improve the OOD F1 score by a large margin.

With large scale data and distribution-based split, the controllability of our dataset is good. We demonstrate how to evaluate the compositional generalization ability of models on different training size and different level of distribution shift. We first find an inverse-scaling phenomenon between OOD performance and data scale. According to Kumar et al. (2022), on large distribution shift, fine tuning can distort pretrained features and underperform OOD. We control the level of distribution shift, and empirically prove that the inverse-scaling phenomenon between OOD performance and data scale is caused by large distribution shift.

Our contribution is three-fold: 1) We propose a challenging compositional generalization task, Dish Name Recognition (DINER), and create a large-scale realistic dataset. 2) We develop strong baseline: T5 with continual pretraining and fine tuning with compositional prompting and GPT-3.5 with selective demonstrations. 3) We demonstrate the controllability of our dataset and provide in-

[1]A dish refers to food prepared in a particular way, and dish name is the conventional designation used by people to refer to that particular prepared food.

| Category | # of Dishes | Base Dishes | Target Dish |
|---|---|---|---|
| *CombTwo* | 43 | Beef, Bun | Beef Bun |
| *AddOne* | 11 | Noodles with Gravy, Steamed Perch | Steamed Noodles with Gravy |
| *RecoTwo* | 619 | Braised Pork, Marinated Beef Brisket | Braised Beef Brisket |
| *RecoThree* | 21 | Spicy Rice Noodle, Pork with Mushroom, Braised Chicken | Rice Noodle with Mushrooms and Chicken |

Table 2: Categories of generalization forms in TMCD splits

sights into compositional generalization in the context of dish name recognition.

## 2 Related Work

Compositional generalization is the ability to understand and create new combinations of elements and structures that have been seen before. To measure the compositional generalization ability of models, a series of studies have proposed synthetically-generated datasets based on semantic parsing tasks (Lake and Baroni, 2017; Bastings et al., 2018; Keysers et al., 2020; Kim and Linzen, 2020). However, the synthetic data lack natural language variation (Shaw et al., 2021) such as anaphora, omission and ambiguity (Keysers et al., 2020).

Recently, some works has proposed to evaluate the compositional generalization ability of the model on realistic data. Shaw et al. (2021) proposed two non-synthetic semantic parsing datasets, GEOQUERY and SPIDER-SSP, and divided the data based on template, length and TMCD, but there are only hundreds or thousands of samples in these two datasets. Yin et al. (2021) proposed a larger scale semantic parsing dataset SMCALFLOW-CS, but this dataset only involves a compositional generalization from skills involving a single domain to skills involving two domains. Liu et al. (2022) proposed a counterfactual recipe generation task based on different combinations of ingredients, flavors and other components in a recipe, and examined the model's ability to modify a base recipe according to the change of an ingredient. However, this task only involved 2500 pieces of data and two forms of compositional generalization.

Inspired by Liu et al. (2022), we also study compositional generalization based on the three components of food, action and flavor in recipes. Our dataset is based on large-scale real-world recipe data generation. Compared with synthetic dataset, our dataset contains rich linguistic phenomena and combination forms. Our data splits result in four types of compositional generalization forms. Comparing to exsiting compositional generalization

datasets, our dataset has the advantage of data size and diversity of compostional generalization forms.

## 3 Dataset

### 3.1 Data Preparation

We build DINER dataset on large Chinese recipe corpus XIACHUFANG proposed by Liu et al. (2022), which consists of 1,550,151 real-world recipes from xiachufang.com. To adapt XIACHUFANG for dish name recognition task, we first collect *(instruction, dish name)* pairs from this corpus.

To collect *(instruction, dish name)* pairs, we face the challenge of identifying the actual dish name associated with each recipe. This is not straightforward as XIACHUFANG provides a title that may not necessarily correspond to the dish name, for example, some titles include irrelevant descriptors and emojis. Based on the heuristic that frequently appearing titles stand a good chance to be a conventional dish name, we first clean the original titles by removing non-Chinese characters and content inside the parentheses, then keep the titles of 20 or more occurrences as dish names of their respective recipes. Liu et al. (2022) proposed to map the title to a list of common dishes, we did not adopt this mapping method to avoid possible matching error. To enhance the quality of the recipes associated with each dish, we additionally eliminate any duplicates or recipes containing fewer than 20 characters.

### 3.2 Dish Name Parsing

Chinese dish names are typically composed of food, actions and flavors (Yu and Huai, 2011; An, 2016), resulting in a diverse range of combinations of these components. We build a dish name parser to parse a dish name to a set of components, so that we can create a train-test split for evaluating compositional generalization based on various combinations of components. Existing off-the-shelf Chinese word segmentation models are inadequate for parsing dish names due to the presence of multi-

component words. For example, *braised pork(*红烧肉*)* is typically segmented as a single word, even though it consists of two distinct components: action *braised(*红烧*)* and food *pork(*肉*)*. To this end, our parser uses maximal matching approach, with glossary of common food, actions and flavors constructed as follows:

**Food**   Most of the food components in dish names are staple ingredients (e.g. pork, tofu) in the recipes. We collect such ingredients from the ingredient list of the recipe. There are also some food components indicating the form of the dish (e.g. pizza, bagel), which can be gathered by extracting the nouns in dish names.

**Actions**   The action components in dish names primarily refer to cooking techniques, such as frying, stir-frying, and toasting. We extract the verbs from dish names to build the glossary of actions.

**Flavors**   The flavor components are the taste expression of a dish, e.g., sour and spicy. We extract the adjectives from dish names to build the glossary of actions.

**Glossary Cleaning and Clustering**   The glossary collected in this manner contains many undesired components. To clean the glossary, we employ a filtering process that removes components with fewer than 5 occurrences in dish names. Additionally, we manually check the glossary and eliminate multi-component words, resulting in 535 food items, 40 actions and 35 flavors. We also notice that some of the food items and actions are different mentions of the same food or action, e.g. 凤爪 and 鸡爪 both refer to *chicken feet*. For fair evaluation and preventing data leakage in data splits, we cluster such food items or actions into on class, such that the components in one class are equivalent during evaluation and generating data splits. The clustering results in 31 action classes and 413 food classes.

### 3.3   Distribution-based Data Splits

Most of the existing evaluations targeting compositional generalization split data based on input/output patterns or length. Keysers et al. (2020) proposed to evaluate compositional generalization based on distribution, and generate data splits through *Maximum Compound Divergence (MCD)* approach. MCD requires that both input and output can be parsed into structured representations like trees or graphs. However, the parsing of recipe in-

structions remains challenging, especially for Chinese ones (Liu et al., 2022). Therefore, in our approach, we utilize *Target Maximum Compound Divergence (TMCD)* (Shaw et al., 2021) to create data splits, as it relies on the structure of outputs. TMCD split data by maximizing the divergence between compound distribution of training and test sets, while constrain the components in test set to appear at least once in the training set.

We define compounds as subsets of the component set in dish names. For instance, in the dish name *scrambled eggs with Chinese chives(*韭菜炒鸡蛋*)* with three components *scramble(*炒*)*, *eggs(*鸡蛋*)* and *Chinese chives(*韭菜*)*, the subset *scrambled eggs(*炒鸡蛋*)* is a compound. We employ the greedy algorithm introduced by Shaw et al. (2021) to generate splits that approximately maximize compound divergence.

We study the construction of target dish names in the test set based on the observed components. As revealed in Table 2, there are four categories of compositional generalization forms in TMCD splits:

- *CombTwo*:   Combine two base single-component dish names into a target dish name.
- *AddOne*: Add a component from a base dish name to another base dish name.
- *RecoTwo*: Recombine components from two base dish names into a target dish name.
- *RecoThree*:   Recombine components from three base dish names into a target dish name.

Comparing to other approach to generate data splits, our approach provide more controllability so that we can evaluate compositional generalization on different level of distributional shift (measured by compound divergence). To generate data splits of different compound divergence levels, we early stop the greedy algorithm at different times. We demonstrate how to evaluate compositional generalization on different levels of distributional shift in §6.

### 3.4   Statistics

**Basic Statistics**   We collect 3,803 dish names that can be parsed into these components. There are 223,590 recipes associated with these dish names, which averages to approximately 58.8 recipes per dish. Each recipe is of 180 characters and 8.3 steps on average. This extensive collection of data offers a wide range of combinations. There are 20 forms

| Combination Forms | # of Dishes | # of Recipes | Example |
|---|---|---|---|
| 2 ingredients | 1,080 | 64,421 | Lettuce with Oyster Sauce (蚝油生菜) |
| 3 ingredients | 666 | 35,310 | Congee with Minced Pork and Preserved Egg (皮蛋瘦肉粥) |
| action + 2 ingredients | 635 | 36,311 | scrambled eggs with Chinese chives (韭菜炒鸡蛋) |
| action + ingredient | 576 | 40,776 | stir-fried pork (小炒肉) |
| flavor + ingredient | 251 | 16,822 | spicy chicken feet (麻辣鸡爪) |

Table 3: Top 5 combination forms in DINER Dataset.

| | Training | Test |
|---|---|---|
| # of Dishes | 3,109 | 694 |
| # of Recipes | 187,520 | 37,861 |
| # of Components | 479 | 383 |
| # of Compound | 2,317 | 616 |

Table 4: Statistics of TMCD training and test set.

of combinations, and we illustrate the most popular combinations in Table 3.

**TMCD Splits**  We divide the dish names into a training set and a test set at an approximate ratio of 4.5:1. Table 4 show the basic statistics of TMCD training and test set. The compound divergence of our TMCD splits is 0.98, and only 0.67% of the test set compounds are present in the training set. As for component occurrence, components in the test set appear 19.3 times on average in the training set. For comparison, we also generate random splits as in-distribution setting, where the *(instruction, dish name)* pairs are divided into training set and test set randomly.

### 3.5 Pilot Study

We conduct a pilot study focused on linguistic phenomena, aiming to analyze their impact on task complexity:

**Anaphora**  Recipe text contains rich anaphoric phenomena (Fang et al., 2022). For example, in the recipe of *marinated beef brisket* in Figure 1, *beef brisket* is also mentioned as *meat pieces*. We analyze 50 sampled recipes and find that each recipe has 1.9 anaphoric references of the staple ingredient on average. This requires models to bridge the different mentions to decide the staple ingredient instead of simply counting the presences.

**Omission**  Omission is a prevalent phenomenon observed in Chinese (Huang, 1998), particularly in informal written contexts such as recipes. As shown in Figure 1, the action *stir fry well (*翻炒均匀*)* in recipe *marinated beef brisket* omits its object. In each sampled recipe, there are 2.8 actions where the ingredient is omitted, which add to the difficulty

of modeling the process of cooking and abstract the cooking skills and flavors.

**Ambiguity**  Ambiguity occurs in components of dish names. For example, the word 卤 means the action *marinate* in the recipe *marinated beef brisket (*卤牛腩*)* (Figure 1), while it refers to *gravy*, a food component in another dish *Noodles with Gravy (*打卤面*)*. Among the dish names in our TMCD test set, we find 10 of them with ambiguous components. Such ambiguity makes it more challenging for models to summarize or abstract the components.

### 3.6 Test Set Annotation

In-context learning (ICL) along with large language models (LLMs) has demonstrated impressive performance on both synthetic and non-synthetic compositional generalization datasets (Levy et al., 2022; An et al., 2023). However, because of the large data scale, assessing the entire test set on LLMs such as the GPT series incurs significant costs. Therefore, we propose to sample a high quality subset for LLM evaluation.

Although we have made efforts to clean the recipe instructions, there are still instances where the information is incomplete or contains too much irrelevant expressions, which means random sampling may result in an undesired test set for LLM evaluation. To mitigate this issue, we ask human experts to handpick a high quality recipe instruction for each dish name in test set. The selected recipe instructions must furnish sufficient information to deduce the dish name and avoid excessive irrelevant content. This ensures that the samples in the test set provide sufficient information for dish name recognition and minimize the presence of irrelevant expressions.

### 4 Baseline Methods

### 4.1 Fine Tuning T5

**End-to-end Fine Tuning**  T5 (Raffel et al., 2020) is a strong baseline model for natural language understanding (NLU) tasks like text summarization

and question answering. Some of these NLU capabilities are highly relevant to the dish name recognition task. We first build a baseline through fine tuning pretrained T5 in a seq2seq manner, where the input is *<Instructions> What is the dish name of this recipe?* and the output is the dish name.

**Fine Tuning with Compositional Prompting**
For compositional generalization tasks, it is a good practice to enhance the primary task by addressing the sub-tasks (Corona et al., 2021; Bursztyn et al., 2022). In the context of dish name prediction, there are three sub-tasks: recognizing the food / action / flavor components. Based on this idea, we propose fine tuning with compositional prompting (CP-FT) to enhance compositional generalization in our task.

We first train an auxiliary model to address three sub-tasks in a seq2seq manner. For the action and flavor components, the input is *<Instructions> What is the main action / flavor of this recipe?* and the output is the action / flavor, while for the food components, the input is *<Instructions> What are the staple ingredients or food form of this recipe?* and the output is the list of food components. If some of the components are missing, the output would be an empty string. Prompting is a good method to inject external knowledge or information to language models (Li et al., 2022). To enhance our primary task, we add the auxiliary model's predictions to original inputs as prompts during both the training and inference phases.

**Continue Pretraining**  Continual pretraining is beneficial for domain adaptation (Wu et al., 2021; Kim et al., 2022). To better understand the recipe text and learn culinary knowledge about food, actions and flavors, we continue pretrain the T5 model on the remaining XIACHUFANG corpus. We selectively retain recipes that have titles unrelated to any dish names in our dataset in order to avoid potential data leakage.

### 4.2 In-context Learning of LLMs

LLMs like GPT-3.5 with in-context learning has proven to be highly effective across a wide range of tasks (Logan IV et al., 2021; Bragg et al., 2021). However, in compositional generalization tasks, LLMs fail to generalize well under OOD settings (Hosseini et al., 2022; Qiu et al., 2022). Some principles for demonstration selection like similarity and diversity have been proposed to improve compositional generalization performance (Levy et al.,

---

*(Demonstrations)*
Input: "Slice the lotus root thinly…" What is the dish name of this recipe?
Output: hot and sour lotus root slices (酸辣藕片)

Input: "Slice the gizzards, heart and liver…" What is the dish name of this recipe?
Output: chicken giblets with pickled pepper (泡椒鸡杂)

*(Target recipe)*
Input: "Cut green pepper, red pepper and pickled pepper into small pieces…" What is the dish name of this recipe?
Output:

---

Golden: hot and sour chicken giblets(酸辣鸡杂)
GPT-3.5 Output: Hot and sour chicken giblets stir-fried with green pepper, red pepper and pickled pepper (酸辣鸡杂炒青椒红椒泡椒)

---

Figure 2: An example of selective demonstrations. GPT-3.5 learns the flavor *hot and sour* from demonstrations, but fails to figure out the staple ingredient.

2022; An et al., 2023). Inspired by these principles, for a test sample *(target instruction, target dish name)*, we first retrieve diverse dish names with highest similarity to *target dish name* from the training set. Then for each retrieved dish name, we find the instruction with highest similarity to *target instruction*. We use the retrieved exemplars as demonstrations for LLMs, Figure 2 shows an example of this procedure.

## 5  Evaluation

### 5.1  Evaluation Metrics

Previous works often use exact match accuracy as evaluation metrics (Lake and Baroni, 2017; Bastings et al., 2018; Keysers et al., 2020; Kim and Linzen, 2020). Nevertheless, such metric is too coarse for dish name recognition task, as it may overlook the distinctions between good and bad predictions. For example, given a golden dish name *steamed noodles with gravy*, a prediction *steamed noodles* is better than *steamed dumplings*, but exact match accuracy cannot reflect such difference.

To measure the model performance more accurately, we first parse the prediction into components, then compute micro F1-score of predicted components given the golden components. To deal with those predictions that are unable to parse, suppose the golden component set is $\{c_1, \ldots, c_n\}$, and the prediction is $p$, we define the truth positive count to be $TP = \sum_{i=1}^{n} \mathbb{I}[c_i \in p]$ and approximately count the number of components in $p$ as the length of $p$ divided by the mean length of golden components $n_p = n \operatorname{len}(p) / \sum_{i=1}^{n} \operatorname{len}(c_i)$

Thus we can calculate $precision = TP/n_p$ and $recall = TP/n$.

## 5.2 Experimental Settings and Hyper-Parameters

For T5-based methods, we use a T5-base model pretrained on large Chinese corpus (Zhang et al., 2021) as base model. We use a batch size of 128 and a learning rate of $5e^{-5}$. We split 20% of the training samples as validation set and stop training when validation loss stop decreasing. The Seq2seq training takes about 4 hours on an A40 GPU. For continual pretraining, we collect about 500k recipes and it takes about 50 hours on an A40 GPU.

For LLM-base methods, we use the gpt-3.5-turbo model and evaluate it on the annotated test set. In examplar retrieval, we use F1-score as the similarity score for dish names, and use BM25 to measure the similarity between recipe instructions. We only retrieve 2 examplars due to the lengthy text. To improve the diversity of demonstrations, after the retrieval of first dish name, we will use the uncovered components to retrieve the second examplar. It takes about 2 dollars for API call.

## 5.3 Results

**T5-based methods** The evaluation results of T5-based methods are shown in Table 5. All the approaches exhibit imperfect performance on the random splits, proving that DINER task is challenging even under in-distribution setting. Comparing the results under TMCD splits and random splits, we find that seq2seq fine tuned T5 model fails to generalize to TMCD splits, and suffers from a severe ID-OOD performance drop: 31.6% in F1-score and 44.7% in exact match accuracy.

Continue pretraining enhance the performance of T5 both on TMCD splits and random splits, and the performace gain is larger on TMCD splits (5.0% in F1) comparing to random splits (1.9% in F1). This may be due to the enhance on comprehension of recipe text through continuous pretraining, enabling the model to overcome some of the challenges associated with understanding unfamiliar recipes in the TMCD test set.

The proposed CP-FT method improves the F1-score of the original T5 and continue pretrained T5 by 2.3% and 3.4%, respectively, while achieve comparable performance on random splits. This means that the process of solving sub-tasks and enhancing the primary task with the help of sub-tasks is an effective approach for compositional

| | TMCD | | Random | |
|---|---|---|---|---|
| | F1 | EM | F1 | EM |
| **T5** | 50.8 | 12.4 | 82.4 | 57.1 |
| +CP-FT | 53.1 | 14.2 | 82.3 | 56.5 |
| +Pretrain | 55.8 | 18.4 | 84.3 | **60.3** |
| +CP-FT & Pretrain | **59.2** | 23.2 | **84.4** | 60.2 |
| **gpt-3.5-turbo** | 52.9 | **23.5** | - | - |

Table 5: F1-score and exact match accuracy of T5-based methods under TMCD splits (OOD setting) and random splits (ID setting).

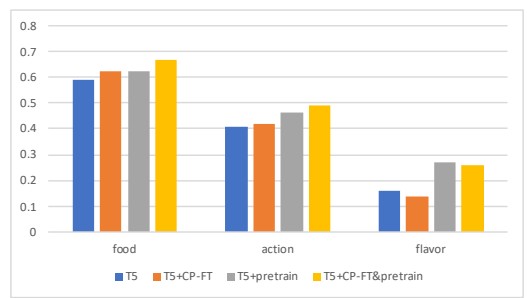

Figure 3: F1-score on TMCD splits of each kind of components. When calculating the F1 score of one kind of components, the samples whose dish name doesn't have this kind of components will be ignored

generalization.

**LLM-based methods** GPT-3.5 with selective demonstrations achieves a F1-score of 52.9% and an exact match accuracy of 23.5% on our annotated test set. In comparison, T5 with continue pretraining and CP-FT achieves a F1-score of 59.4% and an exact match accuracy of 22.9% on the same test set. GPT-3.5 fails to generalize on TMCD splits, and fall behind the fine tuned small language models like T5 in F1-score. GPT-3.5 get higher exact match accuracy than T5-based approach, which is possibly because it successfully recognizes a small part of the recipes similar to observed recipes from its training data, but fails to generalize to a larger part of unfamiliar recipes.

## 5.4 Analysis

**Components** We analyze the model's performance of recognizing each kind of components on TMCD splits. As shown in Figure 3, the food components are the easiest to recognize, and the flavor components are the hardest. Among the T5-based methods, continue pretraining enhance the performance on all three kinds of components, indicating that pretraining on recipe text generally improve model's ability to understanding recipe text. The CP-FT approach enhances the performance on food components and action components, while

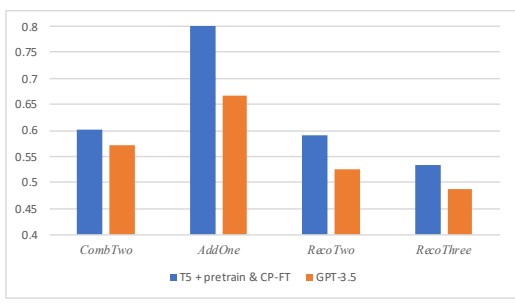

Figure 4: F1-score on TMCD splits of each category of generalization forms.

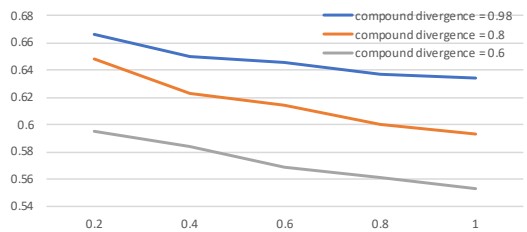

Figure 5: The F1-score of continue pretrained T5 model under different data scales and different levels of distributional shift.

underperforms on flavor components. The undesired performance on flavor components might be attributed to the unbalanced distribution of samples. Among the training data, only 12.8% of the samples have flavor components, this may cause the auxiliary model to predict more empty strings, and mislead the primary task.

**Generalization Forms** We calculate the F1-score of GPT-3.5 and T5 with continue pretraining and CP-FT on the annotated test set, and show the result in Figure 4. Model do surprisingly well on *AddOne*, however, we analyze model's predictions on *AddOne* and find that most of the predictions are the dish name from training set, which is short of one component than the golden dish name. In fact, the exact match accuracy of both models on *AddOne* is merely 9.1%. This means model overfits the training data and tend to generate observed dish names. Among the other three categories, both models performs best on *CombTwo*, and worst on *RecoThree*, which agrees with our intuition on the complexity of these two categories of generalization forms.

## 6 Demonstrations of Data Controllability

With the large data scale and distribution-based data splits, the DINER dataset has good controllability, which enable us to study the model's performance under different data scales and different levels of distributional shift. In this section, we demonstrate such controllability by evaluating the continue pretrained model under multiple conditions.

To generate data splits of different levels of distributional shift, we early stop the splitting algorithm and generate another two splits with compound divergence of 0.8 and 0.6. For each data splits, we sample different proportion $p$ of training data for

training ($p \in \{0.2, 0.4, 0.6, 0.8, 1.0\}$). We evaluate the continue pretrained T5 model on three data splits (compound divergence $\in 0.98, 0.8, 0.6$) and five different $p$s.

As revealed in Figure 5, under all the three data splits, there is an inverse-scaling phenomenon between OOD performance and data scale, which is opposite to the usual case when the training data and test data are sampled from the same distribution. According to Kumar et al. (2022), on large distribution shift, fine tuning can distort pretrained features and underperform OOD. We propose that the inverse-scaling phenomenon between OOD performance and data scale is caused by large distribution shift. We empirically verify this by comparing model's performance under different data splits. As the compound divergence decrease, the downward trend ease, indicating that the degree of inverse-scaling has a positive correlation with the level of distributional shift.

## 7 Conclusion

We propose DIsh NamE Recognition (DINER) task to evaluate natural language understanding ability under compositional generalization settings, and create a large-scale realistic dataset with rich combinations and generalization forms. We provide two strong baselines: 1) T5 with continual pretraining and fine tuning with compositional prompting; 2) GPT-3.5 with selective demonstrations. Our T5-based approach outperforms the plain seq2seq trained version by a large margin (8.4%). However, the ID-OOD performance gap remains unsolved. We also demonstrate the controllability of our dataset and provide insights into compositional generalization in the context of dish name recognition. By proposing this task, we provide a realistic and flexible benchmark for evaluating generalization.

## Limitations

**Limitation of dish name parser.** Our dish name parser is based on maximal matching. While this method has proven efficient in some instances, it does come with inherent limitations such as ambiguity issues. For example, if a dish name can be segmented in multiple ways, the parser may struggle to determine the correct segmentation.

**Imperfection of current solutions.** While our proposed methods offer a strong baseline for compositional generalization in the task of dish name recognition, they do not fully address the issue. Given the challenges outlined, we aim to develop more effective solutions in future.

**Scalability to other languages.** Although our initial investigation centered around Chinese dishes, we acknowledge the importance of validating our conclusions across different languages. For instance, in English, dish names such as Spicy Fried Chicken and Mashed Sweet Potatoes are also composed of ingredients, flavors, and actions. Additionally, there are extensive recipe databases like recipe1M+ available, which could be utilized to create a compositional generalization dataset for dish name recognition in English. Therefore, we recognize the potential to extend our research to other languages and will consider this as part of our future endeavors.

## Acknowledgments

This work is supported by NSFC (62161160339). We would like to thank the anonymous reviewers for the helpful suggestions, and our great annotators for their careful work. Also, we should thank Quzhe Huang and Chen Zhang for their detailed comments. For any correspondence, please contact Yansong Feng.

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
