# OpenReview forum: "DiNeR: A Large Realistic Dataset for Evaluating Compositional Generalization"
_EMNLP/2023/Conference — EMNLP 2023 Main_

### Official Review · Reviewer_A6aj · 2023-08-02

**Soundness:** 4

**Excitement:**

3: Ambivalent: It has merits (e.g., it reports state-of-the-art results, the idea is nice), but there are key weaknesses (e.g., it describes incremental work), and it can significantly benefit from another round of revision. However, I won't object to accepting it if my co-reviewers champion it.

**Paper Topic And Main Contributions:**

The paper proposes a new dataset (DiNeR) for compositional generalization. The dataset consists of dish recipes with the target being the name of the dish, which is compositional of food, actions and flavors descriptors. DiNeR is in Chinese, features natural language and has a large size. The authors provides TMCD splits with different compositional generalization categories and experimental results with sequence to sequence models.

**Questions For The Authors:**

A) Why is the f1 metric preferred to exact match (also in view of what written above)?

B) In the dataset preparation how much were human involved (cleaning the data, the labels ...)? It is not clear from the description whether it was mostly automatic work and it would be interesting to know/expand on it

**Reasons To Accept:**

- The paper introduces a new large dataset for compositional generalization in natural language and with different generalization categories.

- The authors presents an experimental setup with some strong baseline methods. The results shows that the compositional splits are indeed hard w.r.t. to the iid one.

- The paper is clear and well written.

**Reasons To Reject:**

- In the experimental part, the f1 evaluation metric might not always be well suited. For example, in the AddOne category it should be easy for the model to "Classify" the parts already seen in training, which the metric rewards, while the difficult part is the generalization in the new component.

**Reproducibility:**

3: Could reproduce the results with some difficulty. The settings of parameters are underspecified or subjectively determined; the training/evaluation data are not widely available.

**Reviewer Confidence:**

4: Quite sure. I tried to check the important points carefully. It's unlikely, though conceivable, that I missed something that should affect my ratings.

**Typos Grammar Style And Presentation Improvements:**

- Consider adding a statement that the dataset is in Chinese in the introduction or even in the abstract if possible, so that it is immediately clear to the reader.

- While there are some references in the introduction, please consider adding a comprehensive related work section, especially for compositional generalization since there are many works on it currently.

---

> ### Author Rebuttal · Authors · 2023-08-28
>
> Dear Reviewer A6aj,
>
> We appreciate the thorough review of our paper and the insightful feedback provided by the reviewer. We would like to address the questions and concerns raised by the reviewer as follows:
>
> **A) Choice of Evaluation Metric**
>
> > In the experimental part, the f1 evaluation metric might not always be well suited. For example, in the AddOne category it should be easy for the model to "Classify" the parts already seen in training, which the metric rewards, while the difficult part is the generalization in the new component.
>
> We understand the point raised about the AddOne category and how the F1 metric might favor parts seen in training. Actually, we have pointed out that F1 metric overestimates model's compositional generalization ability on the AddOne category at line 519.
>
> To address this issue, we will include exact match metric when analyzing the generalization categories in our revision. For the whole test set and annotated test set, the exact match of models on AddOne category are as follows: (GPT-3.5 was only evaluated on the annotated test set)
>
> | test set              | T5   | T5+CP-FT | T5+Pretrain | T5+CP-FT&Pretrain | GPT-3.5 |
> | --------------------- | ---- | -------- | ----------- | ----------------- | ------- |
> | whole(756 samples)    | 11.1 | 12.3     | 15.2        | **29.9**          | -       |
> | annotated(11 samples) | 9.1  | 9.1      | 9.1         | 9.1               | 9.1     |
>
> The exact match score for all models on the annotated test set is 9.1. However, this score might not accurately capture differences among the four methods due to the limited number of AddOne samples available for comparison.
>
> While on the whole test set, the compositional prompt method improve the exact match on AddOne category by a large margin, which is coherent to our findings.
>
> > Why is the f1 metric preferred to exact match
>
> As mentioned at line 414, we favor the F1 metric over exact match (EM) because exact match can overlook nuances between predictions. For instance, given the reference "steamed noodles with gravy" the predictions "steamed noodles" and "steamed dumplings" are both incorrect by EM, but in reality, "steamed noodles" is a closer match. F1 considers precision and recall, capturing such distinctions better. For "steamed noodles," F1 is 0.8 (precision=1, recall=2/3), and for "steamed dumplings," it's 0.4 (precision=0.5, recall=1/3), offering finer evaluation.
>
> **B) Dataset Preparation and Human Involvement**
>
> > In the dataset preparation how much were human involved (cleaning the data, the labels ...)?
>
> The steps of dataset preparation that involves human labor are dish name parsing and test set annotation:
>
> - In dish name parsing, human were involved when checking the glossary of food, actions and flavors (mentioned at line 206)
> - In test set annotation, human experts were asked to handpick a high quality recipe instruction for each dish name in test set, where the recipe instructions must furnish sufficient information to deduce the dish name and avoid excessive irrelevant content
>
> We will provide a more detailed description of the dataset preparation process in our revision to address this concern.
>
> **C) Typos, Grammar, Style, and Presentation**:
>
> We appreciate the suggestion to mention the language of the dataset (Chinese) in the abstract or introduction to provide immediate clarity to readers. We will make sure to include this information prominently.
>
> Furthermore, we acknowledge the recommendation to include a comprehensive related work section, especially focusing on compositional generalization. We will expand the related work section to provide a more thorough overview of existing research in this area.

---

### Official Review · Reviewer_m8Dw · 2023-08-04

**Soundness:** 4

**Excitement:**

4: Strong: This paper deepens the understanding of some phenomenon or lowers the barriers to an existing research direction.

**Paper Topic And Main Contributions:**

The paper proposes a large-scale, realistic dataset (DiNeR) consisting of 3k dishes and 228k recipes to evaluate compositional generalization. In this dataset, the input is a recipe instruction, and the output is the dish name, which requires an understanding of diverse combinations of ingredients, actions, and flavors. The authors proposed compositional prompting and leveraged continual pretraining to improve compositional generalization. The experimental results showed that the proposed method outperforms T5 and GPT-3.5 on the TMCD split, demonstrating stronger compositional generalization capability. The large scale of data also enables fine-grained control over the distributional shift, making it a testbed for diagnosing compositional generalization.

**Reasons To Accept:**

1. The proposed large-scale, realistic dataset is a good resource contribution for compositional generalization. This is important as LMs are becoming more powerful.
2. The proposed compositional prompting shows potential in a relatively small LM, helping it outperform the much larger GPT-3.5.

**Reasons To Reject:**

1. Compared to previous compositional generalization datasets for semantic parsing, the proposed dish name recognition have a narrower downstream application.

**Reproducibility:**

4: Could mostly reproduce the results, but there may be some variation because of sample variance or minor variations in their interpretation of the protocol or method.

**Reviewer Confidence:**

4: Quite sure. I tried to check the important points carefully. It's unlikely, though conceivable, that I missed something that should affect my ratings.

---

> ### Author Rebuttal · Authors · 2023-08-28
>
> Dear Reviewer m8Dw,
>
> We sincerely appreciate your thoughtful review of our paper. We've carefully considered your comments and would like to address the points you raised.
>
> > Compared to previous compositional generalization datasets for semantic parsing, the proposed dish name recognition have a narrower downstream application.
>
> We acknowledge your point about the narrower downstream application of our proposed dish name recognition dataset compared to previous compositional generalization datasets for semantic parsing. While it's true that the application scope might seem more constrained, we believe this specificity offers unique advantages. This focused task provides a foundation for evaluating compositional generalization in a domain that mirrors real-world scenarios, such as recipe recommendation systems, culinary AI assistants, and menu generation. Therefore, even though the application may appear narrower, its relevance to practical contexts should not be underestimated.
>
> Once again, we sincerely appreciate your time and attention to our paper. Your insightful review has enriched our perspective, and we're dedicated to addressing the concerns you've raised to improve the overall quality of our work.

---

### Official Review · Reviewer_DC41 · 2023-08-11

**Soundness:** 4

**Excitement:**

3: Ambivalent: It has merits (e.g., it reports state-of-the-art results, the idea is nice), but there are key weaknesses (e.g., it describes incremental work), and it can significantly benefit from another round of revision. However, I won't object to accepting it if my co-reviewers champion it.

**Missing References:**

No missing reference.

**Paper Topic And Main Contributions:**

This paper proposes a novel large-scale dataset and task called dish name recognition associated with the task of compositional generalization which is the ability to generalize to novel combinations of given familiar units.
Specifically, the authors build upon the XIACHUFANG recipe dataset collecting dish names and parse them to collect the units under the form of ingredients, flavours and actions. Loosely speaking, the novel task consists in predicting the name of the dish given the recipe instructions. The authors introduce two different baselines to assess the validity of the proposed method both based on LLM-based models a) T5 + fine-tuning and (b) GPT-3.5 with selective demonstrations.

**Questions For The Authors:**

- See reasons to reject.
- At ln. 340 you mention asking experts to handpick a high quality recipe instruction, but it is a bit difficult to grasp what is a high quality instruction as it is not explained.



**Reasons To Accept:**

- The problem of predicting dish names from recipe text is interesting, especially framing the task under the composional generalization umbrella.
- The propose dataset and the categorization of the task with increasing complexity appear to be valuable.
- The baselines proposed are of interest, the worflow is clear and understanble.

**Reasons To Reject:**

- Some useful statistics for the dataset are missing: compound frequency, atom frequency/occurence or co-occurence.
- A point is unclear, is this a multi-lingual dataset? Are the findings/conclusion on compositional generalization from one language  valid for another one?

**Reproducibility:**

3: Could reproduce the results with some difficulty. The settings of parameters are underspecified or subjectively determined; the training/evaluation data are not widely available.

**Reviewer Confidence:**

4: Quite sure. I tried to check the important points carefully. It's unlikely, though conceivable, that I missed something that should affect my ratings.

**Typos Grammar Style And Presentation Improvements:**

- ln. 040: "to summarizing"
- ln. 095: the acronym TMCD is introducted here but the meaning is explained at ln. 230

---

> ### Author Rebuttal · Authors · 2023-08-28
>
> Dear reviewer DC41,
>
> We appreciate your thorough reviews and insightful feedback. We would like to address your questions and concerns as follows:
>
> **A) Missing statistics for the dataset**
>
> >  Some useful statistics for the dataset are missing: compound frequency, atom frequency/occurence or co-occurence.
>
> We acknowledge the significance of compound and atom frequencies, which were initially omitted from our paper due to space limitations. To address this, we provide aggregated statistics as follows:
>
> |                            | Training Set | Test Set |
> | -------------------------- | ------------ | -------- |
> | number of atoms            | 479          | 383      |
> | average atom frequency     | 0.0021       | 0.0026   |
> | number of compounds        | 2317         | 616      |
> | average compound frequency | 0.0004       | 0.0016   |
>
> As for atom co-occurence, as stated at ln. 235 that the TMCD splits "constrain the components in test set to appear at least once in the training set". Specifically, test set atoms appear 19.3 times in average in the training set.
>
> Moreover, only 0.67% of the test set compounds are present in the training set, highlighting the TMCD's ability to minimize compound co-occurrence.
>
> In our revision, we will provide a comprehensive statistical analysis that includes these aspects.
>
> **B) The language-transferablility of the findings/conclusion**
>
> > A point is unclear, is this a multi-lingual dataset?
>
> The dataset is in Chinese. We apologize for any confusion and will make sure to clarify this by explicitly stating that the dataset is in Chinese within the introduction and abstract in our revision.
>
> > Are the findings/conclusion on compositional generalization from one language valid for another one?
>
> Regarding the applicability of our findings and conclusions on compositional generalization to other languages, although our initial investigation centered around a single language, we acknowledge the importance of validating our conclusions across different languages.
>
> For instance, in English, dish names such as *Spicy Fried Chicken* and *Mashed Sweet Potatoes* are also composed of ingredients, flavors, and actions. Additionally, there are extensive recipe databases like recipe1M+ available, which could be utilized to create a compositional generalization dataset for dish name recognition in English. Therefore, we recognize the potential to extend our research to other languages and will consider this as part of our future endeavors.
>
> **C) "high-quality recipe instruction" in line 340**
>
> > At ln. 340 you mention asking experts to handpick a high quality recipe instruction, but it is a bit difficult to grasp what is a high quality instruction as it is not explained.
>
> The recipe instructions for test set were handpicked to exclude the instances that have incomplete information or contain excessive irrelevant expressions (as mentioned at ln. 335). So the criteria for high-quality recipe instructions are as follows
>
> - The recipe instructions must furnish sufficient information to deduce the dish name.
> - The recipe instructions must avoid excessive irrelevant content, such as informal chat or descriptions of other dishes.
>
> We will include a detailed description of test set annotation in or revision.
>
> **D) Typos Grammar Style And Presentation Improvements**
>
> In our revision, we will add a brief explanation of TMCD when introduce it and correct the grammatical mistake.

---

### Meta-Review · Area_Chair_gvdp · 2023-09-10

**Recommendation:** 4

**Metareview:**

This paper presents a large-scale realistic dataset, DIsh NamE Recognition (DINER), comprising  3k dishes and 228k recipes. This paper also presents the associated task (predicting the name of the dish given the recipe instructions) to evaluate compositional generalization.
Experiments are well conducted and evaluated various models on this task (fine tuning T5, in-context learning LLM).
The article could benefit from the addition of corpus statistics. It would also be good to clarify the fact that it's in Chinese (which doesn't detract from the interest of the corpus) and that the conclusions may not apply.

---

### Decision · Program_Chairs · 2023-10-07

**Decision:**

Accept-Main

**Comment:**

This paper presents a large-scale realistic dataset, DIsh NamE Recognition (DINER), comprising  3k dishes and 228k recipes. This paper also presents the associated task (predicting the name of the dish given the recipe instructions) to evaluate compositional generalization.
Experiments are well conducted and evaluated various models on this task (fine tuning T5, in-context learning LLM).
The article could benefit from the addition of corpus statistics. It would also be good to clarify the fact that it's in Chinese (which doesn't detract from the interest of the corpus) and that the conclusions may not apply.